# Quantifying the Coach–Athlete–Parent (C–A–P) Relationship in Youth Sport: Initial Development of the Positive and Negative Processes in the C–A–P Questionnaire (PNPCAP)

**DOI:** 10.3390/ijerph16214140

**Published:** 2019-10-28

**Authors:** Ausra Lisinskiene, Marc Lochbaum, Emily May, Matt Huml

**Affiliations:** 1Education Academy, Vytautas Magnus University, Kaunas 44248, Lithuania; marc.lochbaum@ttu.edu; 2Department of Kinesiology and Sport Management, Texas Tech University, Lubbock, TX 79409-3011, USA; emily.may@ttu.edu; 3CECH-Human Services, University of Cincinnati, Cincinnati, OH 45221-0068, USA; matt.huml@uc.edu

**Keywords:** questionnaire, coach, athlete, parent, interpersonal relationships

## Abstract

Youth sport participation is valued worldwide. Coaches, parents, and athlete youth routinely interact. These interactions impact youth sport participation. To date, only a 48-item measure exits assessing the overall perception of the coach–athlete–parent relationship with the same question set for coaches, parents, and athletes. However, this 48-item measure has not undergone quantitative development. Hence, we sought to assess these 48 items and to further develop a valid and reliable instrument measuring the coach–athlete–parent relationship. To do so, two studies were conducted. In Study 1, 308 participants completed the existing 48-item measure, resulting in 15 items that were fit into two dimensions, positive and negative group processes. In Study 2, 678 participants completed the 15-item measure. After examining the analyses, 11 items remained to form the Positive and Negative Processes in the Coach–Athlete–Parent Questionnaire (PNPCAP). In summary, the PNPCAP is a valid brief measure for assessing interpersonal relationships among coach–athlete–parents in both team and individual sport contexts. Future research is needed to continue to develop the scale for construct validity as well as translate the scale into multiple languages to determine validity in across countries.

## 1. Introduction

Sport is a global phenomenon. Likewise, physical inactivity, especially at the youth level, is epidemically low. Sport participation is one prominent avenue for physical activity engagement. Getting and keeping youth involved in sport is public health issue. Given that sport requires coaches and parents, they are important to study. The increasing academic interest in understanding the nature of coach and parental involvement in youth sport has been highlighted in many studies [1,2,3,4]. Research has illuminated both positive and negative forms of the sport experience from the perspectives of coaches [5,6,7,8]; parents [3,9]; and youth participants [4,10]. One less understood aspect, however, surrounds the potentially conflicting role of all three participants, namely, the athletic triangle. The athletic triangle consists of the coach, athlete, and parent. The relationships within this triad can have a significant impact on the overall development of the athlete [1].

Motivated participation of coaches, athletes, and parents in youth sports can be viewed as an effective educational system. This system can be described as a continuous process of positive interaction among all three members of the system. However, recent research findings have shown that such a three-dimensional educational system has been underdeveloped and lacks positivism, and its effectiveness is limited. The athletic triangle can influence the young athlete’s decision to participate in sport, to remain in the sport, and to pursue a sporting career. For example, parents play a major role in a child’s athletic development and are members of the athletic triangle [9,11,12,13,14]. Close and solid relationships between children and parents ensure a consistent feeling of security and confidence [15]. Mutual parent–athlete trust, respect, belief, support, cooperation, communication, and understanding are considered among the most important relationship components that contribute to athlete performance success and satisfaction. By contrast, parental pressure is found in sports with parent demonstration of excessively high expectations by criticizing children following competitions, punishing them, and “withdrawing love” when performances do not meet their expectations [16]. Perceptions of these types of behaviors have been associated with negative aspects of children’s sporting experiences, such as fear of failure, precompetitive anxiety, burnout, and dropout.

Regarding the other adult in the athletic triad, the coach is one of the main figures in sports with direct influence on athletes’ participation [8]. Coaches are not only expected to coach technically and tactically but also to coach and help a person to develop physically, emotionally, socially, and cognitively [5].

To better understand the coach–athlete–parent relationship and the athlete decisions in sport, the authors drew upon attachment theory. The theory states that early experiences with primary caregivers (typically parents) influence a child’s future development of close relationships. A central tenet of attachment theory is the notion that early childhood lays the foundations for the development of personality throughout life. A close and secure relationship with parents transforms in other contexts such as, in this case, in the sport [15]. The parent–child experienced relationship model transforms and is a basis of development of other close relationships. In a long-term athlete development in sport, the athlete becomes attached to coach and forms a new type of relationship. Attachment theory helps in understanding coach–athlete–parent interpersonal relationships in the sport environment and is developed in one direction and having one common goal to increase the the benefits of youth sport. Active participation of all three members of the athletic triad—coaches, athletes, and parent—is important and necessary for the positive development of this system. Such a three-dimensional educational system may be more effective if parents are more actively and positively involved in the sport of their children, athletes are not pressed but motivated by the psychologically positive environment, and coaches are given continuous learning possibilities [5]. The harmonization of the athletic triad remains the main challenge of further research on this issue. Therefore, the overarching theme is to increase understanding of coach–athlete–parent interaction in youth sport and interrelationships between them and to gain an understanding of how these relationships will affect the young athlete. More specifically, to evaluate and analyze the psychological and educational benefits coach–athlete–parent interpersonal relationships and to consider what strategies may have been used to strengthen, support, and empower these relationships to maximize the motivation of young athlete and to minimize the drop out of the sporting phenomenon.

As scientific research shows, in the context of any sport (individual or team sport), it is important to evaluate such interpersonal relationships. However, to date, such evaluation is fragmented. For example, the Coach–Athlete Relationship Maintenance Questionnaire (CARM-Q) was developed by Rhind and Jowett [17]. The Coach Created Empowering or Disempowering Coaching Questionnaire (EDMCQ-C) was presented by Appleton et al. [6]. Additionally, Sanders et al. [18] developed a questionnaire which assesses the Parent and Family Adjustment Processes (PAFAS). The evaluation of three elements of the system, namely, coach–athlete–parent (C–A–P) interpersonal relationships in youth sport, is still unclear. The development of a new instrument that evaluates interpersonal relationships between three members of the athletic triad would ensure the overall psychological and educational environment in the youth sport setting. Based on the results, coaches could have valuable information on how to strengthen interpersonal relations. Moreover, sport administrators could influence specific interventions to stabilize the athletic triad. To this end, only a 48-item questionnaire of the C–A–P exists, developed by Lisinskiene et al. [19]. Recent qualitative research involved two studies where 136 research participants participated in qualitative study 1, and the main categories were found. For the second qualitative study, a follow-up phenomenological study design was conducted and involved 30 participants (10 coaches, 10 athletes, and 10 youth sports parents) where in-depth interviews were completed based on qualitative study 1. Based on both qualitative study results, the following three themes emerged: group processes, motivation, and over-involvement. The two qualitative investigations revealed themes and 48 potential questions to be used in developing a C–A–P questionnaire in further quantifying the C–A–P results.

### Summary and Purpose

Studies have shown that the interpersonal relationships among the athletic triad in youth sports are a complex and dynamic phenomenon [3,9,20]. The evaluation of the C–A–P relationship becomes an important aspect in a youth sport setting. As previously mentioned, quantitative evaluation of the C–A–P relationship is fragmented. Lisinskiene et al. [19] qualitatively investigated the C–A–P relationship, resulting in 48 items across a number of dimensions. To date, Lisinskiene et al.’s [19] 48 items have not been subjected to quantitative questionnaire development, leaving the evaluation of the C–A–P system in youth sport in need of more rigorous research. Thus, the purpose of this two-study investigation was to develop a reliable and valid questionnaire for measuring the C–A–P interpersonal relationships in a youth sport setting based on the 48 items from Lisinskiene et al. [19].

## 2. Materials

### 2.1. Scale Development Overview

The current study sought to develop a measure designed to assess C–A–P interpersonal relationships within youth sports. This instrument would be administered to populations of coaches, athletes, and parents within youth sport to better understand the relationship between all three stakeholder groups and provide support with unhealthy relationships. To create this instrument, the study was completed in three steps. First, Lisinskiene et al. [19] conducted two qualitative studies where the main categories of the C–A–P interpersonal relationship remained. Second, an exploratory factor analysis (EFA) was implemented to allow the authors to better comprehend the underlying structure of the C–A–P-based 48 items across 8 potential C–A–P dimensions from Step 1. Third, based on the results of Step 2, a confirmatory factor analysis (CFA) was performed on a new set of participants who completed 23 of the original 48 items to further refine the results and improve validity and reliability. This current study concerns two more studies. The first study is an exploratory factor analysis, and the second study reports on a confirmatory factor analysis. 

### 2.2. Study 1 Method

#### 2.2.1. Participants 

The sample consisted of 308 participants who each completed the 48-item questionnaire. Of the 308 participants, 68 female athletes identified as individual and 7 as team sports participants and 45 male athletes identified as individual and 34 as team sports participants. The athletes ranged in age from 12 to 19 (M age = 15.56, SD = 1.93). For the coaches, 27 females identified as individual and 6 as team sport coaches and 25 males identified as individual and 17 as team sport coaches. The coaches ranged in age from 23 to 69 (M age = 45.94, SD = 8.69). For the parents, 30 females identified as individual and 11 as team sport parents and 21 males identified as individual and 14 as team sport parents. The parents ranged in age from 31 to 54 (M age = 40.24, SD = 5.55).

#### 2.2.2. Procedure

Following the initial development of the 48-item instrument [19], a new group of participants completed the measure. An approved human subject approval from the first author’s university was presented to all participants. Parents of athletes under the age of 18 provided adult consent as well. For the parents and coaches, a web-based survey link was sent by the principal of the respective sport schools that agreed to participate in the research. Once parent approval was obtained, the youth online survey link was distributed to participants. 

#### 2.2.3. Instrument

All items used a five-point Likert scale, varying from totally disagree (1) to totally agree (5). Likert scale was chosen for consistency and the high probability that most participants would have previous exposure to this format. Additional information was collected on the participant’s role (i.e., coach, parent, or athlete), sports experience (i.e., team or individual sport experience), age, and gender. The instrument’s 48-items were reduced to align with the EFA results. The panel of experts scrutinized the EFA findings to interpret the naming of the resulting factors. 

#### 2.2.4. Data Analyses 

The 48-item C–A–P instrument analysis was through EFA for item purification and reduction. The authors completed an EFA using a principal axis factor method with a direct oblimin rotation. Performing an EFA allows the authors to better understand the underlying structure of the instrument and purification of individual items [21]. A principal axis factoring was chosen because of the hypothesized second-order factors within the instrument. A direct oblimin rotation was chosen because of the expected correlation between factors [22]. To ensure sampling adequacy and structure detection, a Kaiser–Meyer–Otkin (KMO) (significant and higher than 0.90) and Bartlett’s test of sphericity (significant and less than 0.05) were performed [23]. To assess the factor loadings from the EFA results, the researchers interpreted a number of results. First, a combination of eigenvalues, scree plot, and parallel analysis were used to identify the extracted factors. Second, Cronbach’s alpha was examined for interpreting the internal consistency reliability of the items (see Table 1).

### 2.3. Study 2 Methods

#### 2.3.1. Participants 

The instrument’s second version (based on EFA results and researcher insights) was disbursed to a target sample of 678 participants. The participants were recruited from seven general sports education schools in Kaunas, and in Vilnius, Lithuania. We also included two individual and team sports federations to have a bigger sample of coaches. Demographic information from participants was collected as well as additional information, such as the athletes’ years of sports club participation. For athletes, 197 females and 250 males completed the measure. The females ranged in age from 11 to 19 (M age = 13.66, SD = 1.59) and in sports clubs experience from 1 to 6 or more years (M year experience = 3.50, SD = 1.84). When participants were asked to choose a sport type, 91 reported as individual and 106 as a team. The males ranged in age from 11 to 18 (M age = 14.05, SD = 1.52) and in sports clubs experience from 1 year to 6 or more years (M year experience = 4.27, SD = 1.90). When completing the section sport type section, 94 indicated individual and 156 indicated team. For the parents, 74 identified themselves as females and 19 as males. The female parents ranged in age from 31 to 58 (M age = 42.37, SD = 5.62) with the sport type split equal as 37 indicated individual and 37 indicated team sport when asked to choose a sport type. The male parents ranged in age from 34 to 59 (M age = 42.57, SD = 6.29). Concerning sport type, 8 indicated individual and 11 indicated team. For the coaches, 20 identified themselves as females and 19 as males. The female coaches ranged in age from 30 to 60 (M age = 43.60, SD = 8.82). Concerning sport type, the split was 7 individual and 13 teams. The male parents ranged in age from 30 to 62 (M age = 41.05, SD = 7.80), with 11 coaching individual and 8 coaching team sports.

#### 2.3.2. Instrument

A second version of the instrument was created with eight additional items that had suitable statistical properties in the EFA (see Table 2). The additional eight items were added for scrutiny for confirmatory factor analysis. This second version hypothesized the simpler conceptual two-factor framework of Positive Group Processes and Negative Group Processes.

#### 2.3.3. Procedure

An approved human subject approval from the first author’s university was presented to all participants. Parents of athletes under the age of 18 provided adult consent as well. For the parents and coaches, a web-based survey link was sent by the principal of the respective sport schools that agreed to participate in the research. Once parent approval was obtained, a youth-specific survey link was distributed to participants. 

#### 2.3.4. Data Analysis

Following EFA results, a second sample was collected for completing the CFA. Performing a CFA allows the researchers to specify the number of factors based on previous EFA results to confirm or reject measurement theory [22]. To ensure the data fit the EFA’s hypothesized structure, model fit was examined. The authors followed Hu and Bentler’s [24] recommendations of model fit: comparative fit index (CFI) of 0.90 or greater, an adjusted goodness-of-fit index (AGFI) of 0.80 or greater, and a root mean square error of approximation (RMSEA) of 0.06 or less. Lastly, factor loadings of 0.50 or greater were retained to ensure minimum standardized large factor loadings [25].

## 3. Results

### 3.1. Study 1 Results

The results from the EFA are provided in Table 1. Pretesting was successful, as KMO was 0.937 and Bartlett’s test was statistically significant *p* < 0.01 [23]. Results show a total of three factors with eigenvalues greater than 1.0 and a total variance explained of 66.85%. Results from the scree plot and parallel analysis also confirmed three factors. There was a wide discrepancy of variance explained across all three factors, with the largest factor (group processes) explaining 38.4% of total variance and the smallest factor (over-involvement) explaining 4.4% of total variance. These results created a reduction of total items from 48 to 15 across three factors (see Table 1 for specific questions): (a) group processes, (b) motivation, (c) over-involvement. Additionally, Cronbach’s alpha for all three factors was strong (ranging from 0.82 to 0.92), above Nunnally and Bernstein’s [26] recommendation of 0.70 or higher. Item factor loadings were above the minimum threshold of 0.40, ranging from 0.65 to 0.83 [27]. One concern was the cross-correlation between items within motivation and some items on group processes.

Further examination of the cross-correlation between factors led the researchers to interpret that the participants were simply viewing group processes from a dichotomous relationship: (a) positive group processes (the group processes and motivation factors) and (b) negative group processes (the over-involvement factor). Because of this interpretation, the instrument was redesigned conceptually to focus in Study 2 on a two-factor solution within group processes.

### 3.2. Study 2 Results

The results of the 23 item C–A–P with initial conceptualized dimension and item number and CFA are provided in Table 2 and Table 3. Goodness-of-fit scores were examined first. The chi-square for the model was statistically significant. Normally, this is an unfortunate finding, but with such a large sample size, this was expected by the researchers and further scrutiny has been recommended only if sample sizes are less than 200. Our model fit scores were above the standard of threshold established by Hu and Bentler [24]: CFI = 0.964, AGFI = 0.940, RMSEA = 0.059), meaning our model fit the predicted data, and further analysis on factor loadings is warranted. All 11 items had factor loadings above Kline’s [25] recommended threshold of 0.40, with factor loadings ranging from 0.53 to 0.77. Because model fit was sufficient, no modification indices or bootstrapping options were considered by the researchers. 

## 4. Discussion

This study aimed to validate the PNPCAP as brief inventory designed to assess the interpersonal relationships between the coach, the athlete, and the parent that could be used in an individual or team sports context. PNPCAP assessed two domains associated with interpersonal relationships and described either positive or negative group processes between the C–A–P members.

This questionnaire development process was framed in a previous qualitative work of Lisinskiene et al. (2019), where the main categories were found relating to interpersonal relationships of the C–A–P. The studies in a similar field [28] developed a questionnaire which evaluates the coach–athlete interpersonal relationships, and researchers presented the Coach–Athlete Relationship Questionnaire (CART-Q) development process. Researchers also grounded the measurement development process in previous qualitative case studies and found the main information for measurement development. The CART-Q questionnaire involved a development process where commitment, closeness, and complementarity described the overall coach–athlete relationship. We followed the study by Lisinskiene et al. [19] which included a 48-item measure assessing the interpersonal relationships between the C–A–P; however, as was mentioned previously, this 48-item measure has not undergone quantitative development. This current study performed a final development process and presented a final version of valid and reliable PNPCAP measurement. In study 1, an exploratory factor analysis was performed (Table 1 and Table 2). In study 2, a confirmatory factor analysis was obtained to refine the results and improve the validity and reliability of the measure (Table 3).

Referring back to study 1, we subjected the resultant scales to rigorous psychometric evaluation and found support for the 15-item, three-factor structure of the C–A–P questionnaire with excellent scale internal consistency. As it is presented in Table 1, out of the 48 items in the original questionnaire, a 15-item scale was created, with 33 items needing to be removed from the C–A–P relationship scale because they had no statistical support, as statistical analysis showed these items were designed to mainly measure the same daily relationships and duplicated each other as well as had the same meaning. This may explain why they did not load significantly on the overall C–A–P scale and needed to be removed. As an outcome of study 1, three factors emerged and were named as group processes, motivation, and over-involvement. Although the first version of the C–A–P scale showed adequate convergent validity and a good scale of internal consistency, researchers analyzed and interpreted the results in more depth. Cross-correlation between items within motivation and some items on group processes were further examined and led the researchers to interpret that the participants were viewing group processes from a dichotomous relationship: (a) positive group processes (the group processes and motivation factors) and (b) negative group processes (the over-involvement factor). The instrument was redesigned conceptually to focus in Study 2 on a two-factor solution within group processes.

We found that the three factors that emerged could be formed as a two-factor scale (positive and negative group processes), as research participants see group processes from a dichotomous relationship. Relating to this, we interpreted that group processes and motivation could be merged because research participants understand the same meaning of it. In addition to this, the factor over-involvement was renamed as negative group processes. In more detail, the PNPCAP is positive where three-way communication, trust, support, teamwork, and respect are present. PNPCAP is negative where at least one member of the C–A–P is over-involved. To re-test the two factors (positive and negative group processes), i.e., the instrument’s second version with a set of 15 questions, we made a rigorous psychometric evaluation again with a bigger sample participants in study 2. In addition to this, we added 8 additional questions from the first version of the scale that had the best suitable fit in the scale to re-test the results again (see the 8 bold items in Table 2). After a careful research and data analysis of study 2 with a set of 23 questions of the measurement, we found that the most suitable and significantly sound results are the 11-item questions that measure the interpersonal relationships among the C–A–P (Table 3). In this sense, the 8 positive and 4 negative group processes between the C–A–P members emerged. The 8 items that were added from the first study (bold items in Table 2) had a good fit in the overall examination of study 2. In study 2, a new group of a bigger sample of research participants was involved, and the re-test analysis was done. The confirmatory factor analysis showed that from the 8 items added from the 23-item C–A–P questionnaire, three of te items numbered 1, 20, and 23 (Table 3) may be included in the questionnaire as they had a good statistical score which could fit to the PNPCAP. In this sense, the researchers created a reasonable solution to add the most suitable and high-scored items from study 1. As a result, the study 2 result revealed that the modified factor structures of the PNPCAP are stable across the samples of coaches, athletes, and parents and supported the stability of the 11-item two factor model of PNPCAP. The final 11-item PNPCAP inventory version is outlined below (Table 4). Psychometric evaluation of the PNPCAP revealed that the scale had good internal consistency, as well as a satisfactory construct and predictive validity. The PNPCAP is a brief 11-item scale, reliable and valid to assess 12–18-year-old adolescent athletes’, their coaches’, and parents’ perceptions about interpersonal relationships between the coach, the athlete, and the parent in a youth sports setting. Potential uses of the measure and implications for future validation studies are discussed. This study provides scientific evidence of the coach–athlete–parent interpersonal relationship and acts as instrumentation to measure the inner C–A–P relationship. 

## 5. Limitations and Perspectives for Future Research

This study has one main limitation. We used convenience samples and thus did not take into account participant characteristics such as potential age and gender differences across coaches, athletes, and parents. The hope of course is the PNPCAP will be found reliable and valid in youth sport. Future research should focus on whether our new instrument is psychometrically sound considering participant characteristics such as gender, age, sport types, parenting styles, and coaching experiences. Beyond reliability and validity, it would be beneficial to separately evaluate the PNPCAP from larger samples of coaches, parents, and athletes to examine important potential correlates. Moreover, the PNPCAP evaluation in different cultures is warranted, as sport is globally valued and practiced. Practically, by identifying what makes an effective and positive coach–athlete–parent relationship, coach and parent education and intervention programs could be initiated by sports club psychologists or sports educators. Such educational or psychological intervention programs could be designed to educate coaches, athletes, and parents and strengthen the athletic triad. In addition, the aforementioned educational or intervention programs could be developed and could be implemented by National Governing Sports Federations. For instance, USA university and such university programs modeling the USA university sports program system could include such topics in their curricula and explain to students the importance of interpersonal relationships in youth sports and educating future young coaches.

## 6. Conclusions

Psychometric evaluation of the PNPCAP revealed that the scale had good internal consistency, as well as satisfactory construct and predictive validity. The PNPCAP is a brief 11-item scale, reliable and valid to assess 12–18-year-old adolescent athletes’, their coaches’ and parents’ perceptions about interpersonal relationships between the coach, the athlete, and the parent in a youth sports setting. Potential uses of the measure and implications for future validation studies are discussed. 

## Figures and Tables

**Table 1 ijerph-16-04140-t001:** Exploratory factor structure and commonalities.

Item	F1	F2	F3
Factor 1 = Group processes			
18. Everyone in my CAP talks honestly.	0.69		
19. In my CAP, everyone cares for one another.	0.72		
20. In my CAP, everyone helps with the tasks required for success.	0.68		
26. My CAP allows open expression of ideas.	0.73		
27. My CAP relationship is reliable during hardship.	0.74		
28. In my CAP, everyone cares for one another.	0.76		
35. My CAP is positive.	0.77		
36. In my CAP, everyone works together.	0.80		
43. My CAP is supportive.	0.73		
Factor 2 = Motivation			
7. Passion to achieve a common goal characterizes my CAP.		0.71	
15. My CAP is enthusiastic.		0.83	
23. My CAP encourages effort.		0.65	
Factor 3 = Over-involvement			
14. In my CAP, at least one member oversteps boundaries.			0.68
30. At least one member in my CAP is too demanding.			0.77
38. At least one member in over-involved.			0.77
Eigenvalues	4.999	1.949	1.945
% of variance explained after rotation	33.33	12.99	12.97

**Table 2 ijerph-16-04140-t002:** The 23-item coach–athlete–parent (C–A–P) with initial conceptualized dimension and item number.

New Item Number	Original Item Number, Question (Category)
**1**	6. In my C–A–P, at least one member expects too much. (Over-involvement)
2	7. Passion to achieve a common goal characterizes my CAP. (Motivation)
**3**	10. Everyone in my C–A–P talks openly. (Communication)
**4**	11. Everyone in my C–A–P helps in both wins and losses. (Support)
5	14. In my C–A–P, at least one member oversteps boundaries. (Over-involvement)
6	15. My C–A–P is enthusiastic. (Motivation)
**7**	17. In my C–A–P, everyone is honest with each other. (Trust)
8	18. Everyone in my C–A–P talks honestly. (Communication)
9	19. In my C–A–P, everyone cares for one another. (Support)
10	20. In my C–A–P, everyone helps with the tasks required for success. (Teamwork)
11	23. My C–A–P encourages effort. (Motivation)
**12**	25. My C–A–P is dependable. (Trust)
13	26. My C–A–P allows open expression of ideas. (Communication)
14	27. My C–A–P relationship is reliable during hardship. (Support)
15	28. In my C–A–P, we are a team. (Teamwork)
16	30. At least one member in my C–A–P is too demanding. (Over-involvement)
**17**	31. Everyone in my C–A–P works hard to achieve a common goal. (Motivation)
18	34. Everyone in my C–A–P listens to each other’s point of view. (Communication)
19	35. My C–A–P is positive. (Support)
**20**	36. In my C–A–P, everyone works together. (Teamwork)
21	37. Mutual respect characterizes my C–A–P. (Respect)
22	38. At least one member is over-involved. (Over-involvement)
**23**	43. My C–A–P is supportive. (Support)

**Note.** Bold indicates items added to 15 exploratory factor analysis (EFA) items in Study 2.

**Table 3 ijerph-16-04140-t003:** Standardized solutions and goodness-of-fit indicators for the C–A–P measures.

Original Question Number, Question (Revised Question Number)	Standardized Values	
	F1	F2
27. My C–A–P relationship is reliable during hardship. (14)	0.69	
28. In my C–A–P, we are a team. (15)	0.66	
35. My C–A–P is positive. (19)	0.75	
36. In my C–A–P, everyone works together. (20)	0.72	
37. Mutual respect characterizes my C–A–P. (21)	0.77	
43. My C–A–P is supportive. (23)	0.68	
11. Everyone in my C–A–P listens to each other’s point of view. (18)	0.64	
6. In my C–A–P, at least one member expects too much. (1)		0.53
14. In my C–A–P, at least one member oversteps boundaries. (5)		0.61
30. At least one member in my C–A–P is too demanding. (16)		0.56
38. At least one member is over-involved. (22)		0.65

**Table 4 ijerph-16-04140-t004:** Final version of the Positive and Negative Processes in the Coach–Athlete–Parent Questionnaire (PNPCAP) interpersonal relationships of the C–A–P.

Item Description	Totally Disagree	Disagree	Neither Agree, Neither Disagree	Agree	Totally Agree
1. My C–A–P relationship is reliable during hardship (P-Support) *	1	2	3	4	5
2. In my C–A–P, we are a team (P-Support) *	1	2	3	4	5
3. My C–A–P is positive (P-Support) *	1	2	3	4	5
4. In my C–A–P, everyone works together (P-Teamwork) *	1	2	3	4	5
5. Mutual respect characterizes my C–A–P (P-Respect) *	1	2	3	4	5
6. My C–A–P is supportive (P-Support) *	1	2	3	4	5
7. Everyone in my C–A–P listens to each other’s point of view (P-Communication) *	1	2	3	4	5
8. In my C–A–P, at least one member expects too much (N-Over-involved) **	1	2	3	4	5
9. In my C–A–P, at least one member oversteps boundaries (N-Over-involved) **	1	2	3	4	5
10. At least one member in my C–A–P is too demanding (N-Over-involved) **	1	2	3	4	5
11. At least one member is over-involved (N-Over-involved) **	1	2	3	4	5

**Note.** The explanation of the item distribution in two higher order factors: * (P-subscale)—Positive group processes; ** (N-subscale)—Negative group processes.

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
