# Peer review of "Quantifying the Coach–Athlete–Parent (C–A–P) Relationship in Youth Sport: Initial Development of the Positive and Negative Processes in the C–A–P Questionnaire (PNPCAP)"

_ijerph, 2019, doi:10.3390/ijerph16214140_

Round 1

Reviewer 1 Report

The manuscript is very interesting and very well described. It will be positive in future approaches to present results between individual and team sports in order to analyse if the CAP relationship in youth sports differs according to the sport practiced.

Author Response

Dear Reviewer, 

Along with my co-authors, I thank you for the opportunity to revise and resubmit our manuscript, Quantifying the Coach-Athlete-Parent (C-A-P) Relationship in Youth Sport: Initial Development of the Positive and Negative Processes in C-A-P Questionnaire (PNPCAP).

The response to your review could be found in the attached file.

Thank you,

Authors

Reviewer 2 Report

This topic is interesting. I have the following comments: - The topic is related to public health. However, it is written in a way that is irrelevant in public health. As this journal is a public health journal, I would like to suggest the authors to connect it's relevance to public health in the introduction part and the discussion part e.g. implications to public health. -The structure of the manuscript may not comply with the journal. Normally, the part Methods and Materials should be written separately from the results part. Besides, in the introduction part, too much is talked about the literature review. It may also be better that the authors to put a subsection of a review of literature in the Methods and Materials part. In this way, the structure of the article will look much better. - The authors didn't explain the details of how to establish the sample size, how to establish the exact number of the three groups and how to balance the gender differences. If this is the limitation of the design, it should be stated in the limitation part. - The limitation part is missing. The authors should write it in the last paragraph of the Discussion part. - About the statistical part, it is suggested to find a statistician for further review.

Author Response

(The authors gave the same response as above.)

Reviewer 3 Report

This paper checks the validity of an instrument that measures the impact on youth sports participation of interactions between coaches, parents and young athletes.

Below are general and specific comments to improve the content and readability of the manuscript.

General comments:

The introduction provides sufficient background on the topic and previews major points. Both research design and analysis are adequate.

Specific comments:

1/35-37: Citation needed. 7/237: It would be necessary to add a brief paragraph about possible limitations of the study.

Author Response

(The authors gave the same response as above.)

Round 2

Reviewer 2 Report

I am just not satisfied with the structure. Every journal has its guidelines for the structure. I suggest to separate the results from the Methods part. For the research in itself it can be accepted now.

Author Response

Dear Reviewer,

Thank you very much for your suggestion. We agree, it was our mistake and we corrected it. 

Thank you,

Authors
